# Random access parallel microscopy

**Mishal Ashraf[1†], Sharika Mohanan[2†], Byu Ri Sim[1†], Anthony Tam[1†], Kiamehr Rahemipour[1], Denis Brousseau[3], Simon Thibault[3], Alexander D Corbett[2]*, Gil Bub[1]***

[1]Department of Physiology, MGill University, Montreal, Canada; [2]Department of Physics and Astronomy, University of Exeter, Exeter, United Kingdom; [3]Department of Physics, Engineering Physics and Optics, Université Laval, Laval, Canada

**Abstract** We introduce a random-access parallel (RAP) imaging modality that uses a novel design inspired by a Newtonian telescope to image multiple spatially separated samples without moving parts or robotics. This scheme enables near-simultaneous image capture of multiple petri dishes and random-access imaging with sub-millisecond switching times at the full resolution of the camera. This enables the RAP system to capture long-duration records from different samples in parallel, which is not possible using conventional automated microscopes. The system is demonstrated by continuously imaging multiple cardiac monolayer and *Caenorhabditis elegans* preparations.

***For correspondence:**
A.Corbett@exeter.ac.uk (ADC);
gil.bub@mcgill.ca (GB)

[†]These authors contributed equally to this work

**Competing interests:** The authors declare that no competing interests exist.






## Introduction

Conventional multi-sample imaging modalities either require movement of the sample to the focal plane of the imaging system (***Klimas et al., 2016***; ***Yemini et al., 2013***; ***Kopljar et al., 2017***; ***Hortigon-Vinagre et al., 2018***), movement of the imaging system itself (***Likitlersuang et al., 2012***; ***Hansen et al., 2010***), or use a wide-field approach to capture several samples in one frame (***Larsch et al., 2013***; ***Taute et al., 2015***). Schemes that move the sample or the imaging system can be mechanically complex and are inherently slow, while wide-field imaging systems have poor light collection efficiency and resolution compared to systems that image a single sample at a given time point. An important limitation of current imaging modalities is that they cannot continuously monitor several samples unless they are in the same field of view. As many experiments require continuous long-term records in spatially separated samples, they cannot benefit from these high-throughput techniques.

The random-access parallel (RAP) system uses a large parabolic reflector and objective lenses positioned at their focal distances above each sample. A fast light-emitting diode (LED) array sequentially illuminates samples to generate images that are captured with a single camera placed at the focal point of the reflector. This optical configuration allows each sample to fill a sensor's field of view. Since each LED illuminates a single sample and LED switch times are very fast, images from spatially separated samples can be captured at rates limited only by the camera's frame rate or the system's ability to store data. RAP enables effectively simultaneous continuous recordings of different samples by switching LEDs at very fast rates. We demonstrate the system in two low-magnification, low-resolution settings using single-element lenses and other easily sourced components.

## Results

Our current prototypes (***Figure 1A***) use fast machine vision complementary metal-oxide semiconductor cameras and commercially available LED arrays controlled by Arduino microcontrollers, which can rapidly switch between LEDs at kHz rates. A single-element plano-convex lens is placed above each sample, so that collimated light is projected to a 100 mm focal length parabolic reflector, which

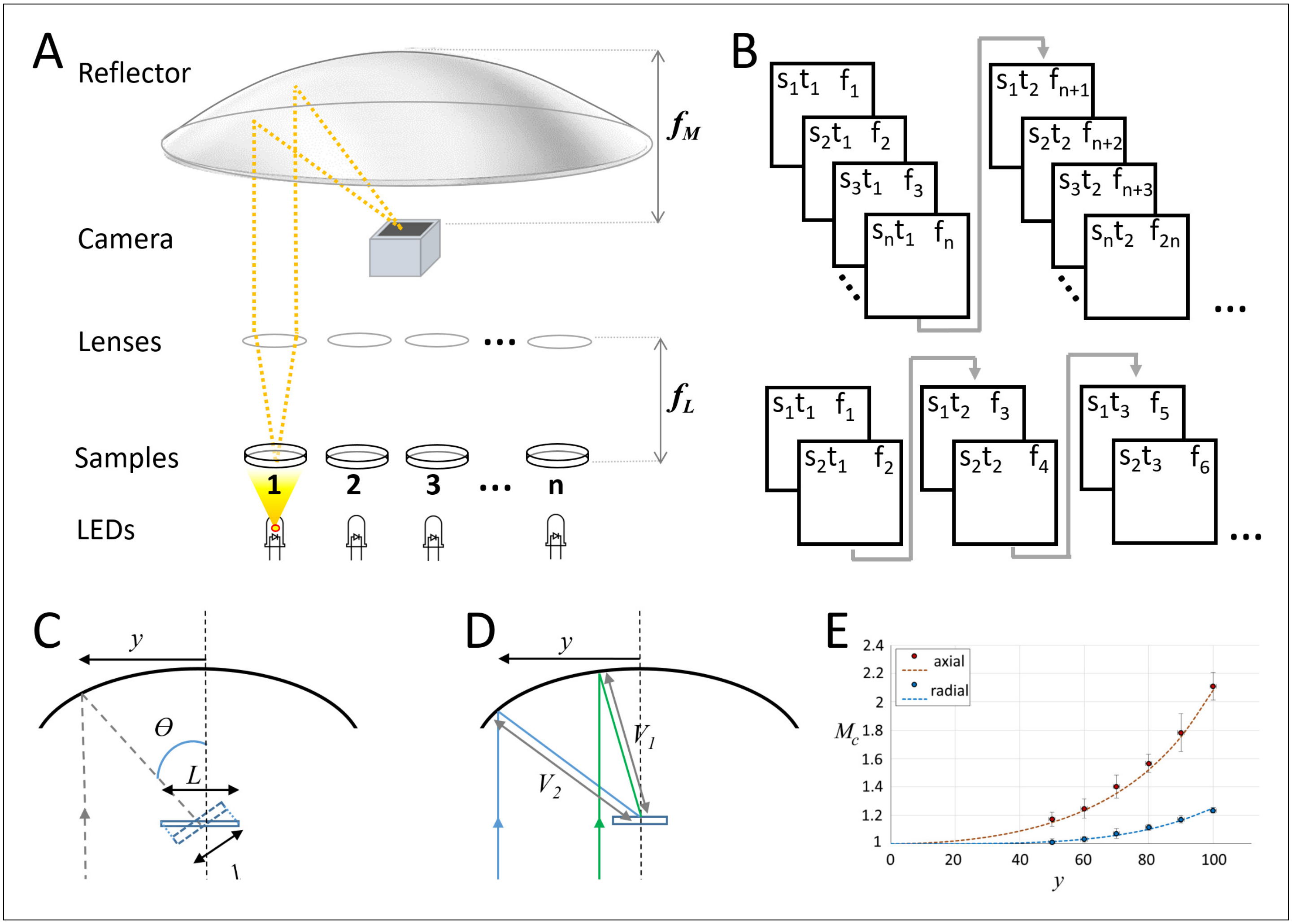


**Figure 1.** Random-access parallel (RAP) imaging principle and magnification properties. (**A**) The random-access imaging system uses a parabolic reflector to image samples directly on a fast machine vision camera located at the focal point of the mirror ($f_M$). Single-element plano-convex lenses are used as objectives, with samples positioned at their focal point ($f_L$). Samples are sequentially illuminated using a LED array controlled by an Arduino microcontroller: a sample is only projected on the sensor when its corresponding LED is 'on'. See ***Figure 1—figure supplement 1*** and ***Table 1*** for details. (**B**) (top) Sample *s*, is captured at time *t*, on frame *f*. For a total of *n* samples, each sample is captured once every *n* frames; (bottom) a smaller subset of samples can be imaged at higher temporal resolution by reducing the number of LEDs activated by the microcontroller. (**C**) Image magnification: the chief ray (dashed line) arrives at the detector plane at an incidence angle θ which increases with lateral displacement, *y*. The image is stretched in the direction parallel to *y* by a factor of *L/l*. (**D**) The image is isotopically magnified as the distance between the mirror and the image increases (*V2*>*V1*) as *y* increases. (**E**) The combined magnification, $M_C$, shows the impact of the combined transformation on the magnification in both image dimensions (*y'* parallel to *y*, and *x'* orthogonal to *y*). Red dots (measured) and dashes (predicted) show magnification in *y'*, and blue dots (measured) and dashes (predicted) show magnification in *x'*, inset shows Images of a grid (200 µm pitch) taken with y = 70 mm, left is the uncorrected image and right shows the correct image using ***Equation 1***.

The online version of this article includes the following figure supplement(s) for figure 1:

**Figure supplement 1.** Configurations used for data collection.

**Figure supplement 2.** Comparison of conventional and RAP images.

then creates an image on the detector. The bright-field nature of the illumination used in this design allows images to be captured with sub-millisecond exposure times. The camera is synchronized with the LED array via a transistor–transistor logic (TTL) signal from the microcontroller, so that a single frame is captured when any LED is on. This setup can rapidly switch to image any dish under the parabolic reflector without moving the sample or camera. In addition, the system can acquire data from several dishes near-simultaneously by trading-off the number of samples for frame rate: for example,

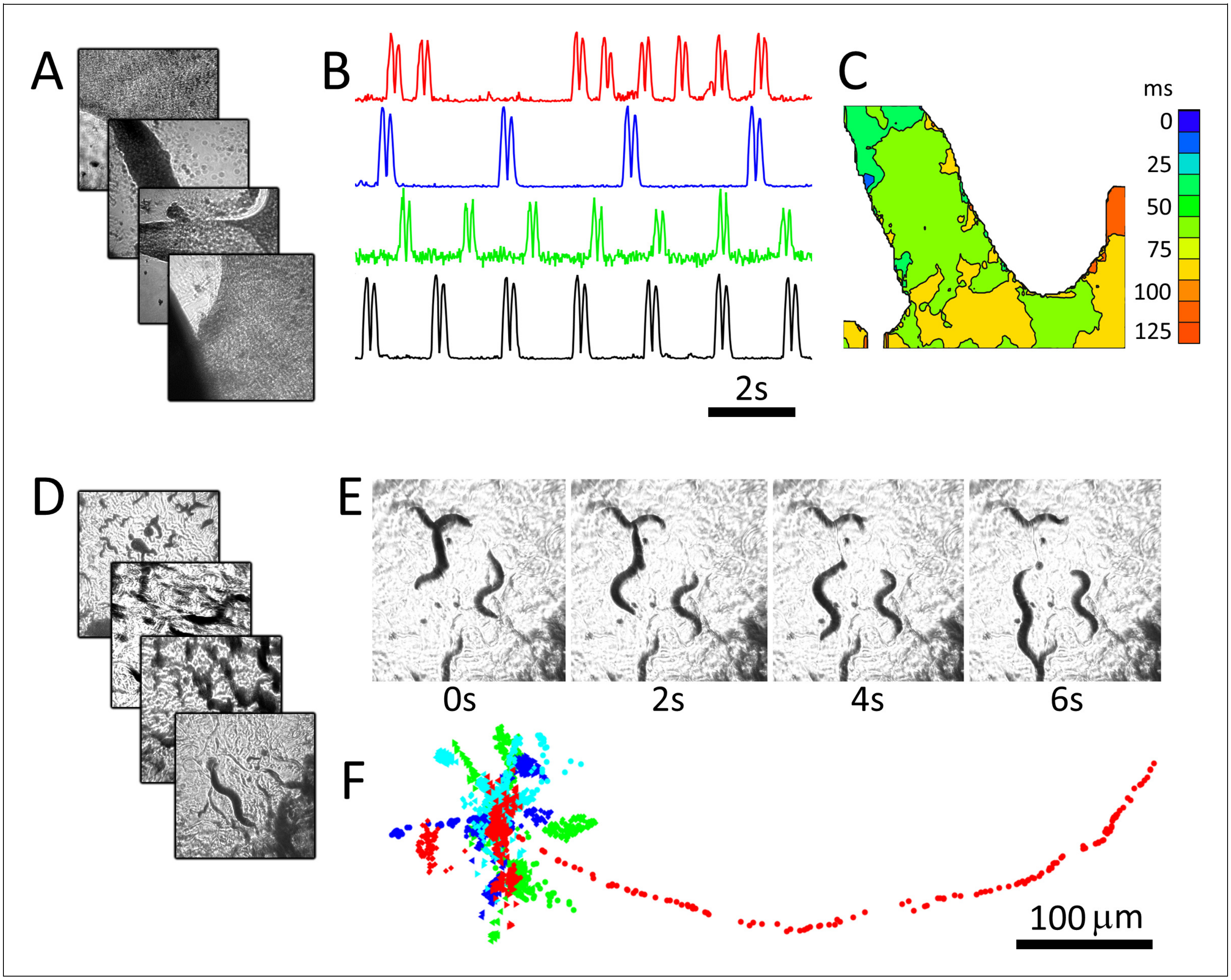


**Figure 2.** RAP imaging of cardiac monolayer and *C. elegans* preparations. (**A**) Four cardiac monolayer preparations in four separate petri dishes are imaged in parallel at 40 fps/dish. (**B**) Activity vs time plots obtained from the four dishes show different temporal dynamics, where double peaks in each trace correspond to contraction and relaxation within a 20 × 20 pixel ROI (see Materials and methods); (**C**) an activation map from the second dish (blue trace in **B**) can be used to determine wave velocity and speed; (**D**) four *C. elegans* dishes imaged in parallel at 15 fps/dish; (**E**) images from one dish every 30 frames (2 s intervals) shows *C. elegans* motion; (**F**) the location of five worms in each dish was tracked from data recorded at 15 fps over 250 frames using open-source wrMTrck (***Nussbaum-Krammer et al., 2015***) software. Dots in different colours (blue, cyan, green, and red) show the tracked positions from plates 1–4, respectively. Each image in (**A**), (**D**), and (**E**) shows a 2 × 2 mm field of view.

if a 500 fps camera is used, 50 dishes can be captured at 10 fps, or any two dishes can be recorded at 250 fps (***Figure 1B***).

The high NA and large field of view offered by parabolic mirrors have made them very attractive to imaging applications beyond the field of astronomy. However, parabolic mirrors introduce off-axis aberrations, which corrupt any widefield image formed (***Rumsey, 1971***; ***Wynne, 1972***). This has resulted in compromises, such as restricting imaging to the focal region and then stage scanning the sample (***Lieb and Meixner, 2001***), which have limited its use to niche applications. In our design, transillumination from LEDs far from the sample and collimation from the objective lens results in mostly collimated light being refocused by the parabolic mirror, avoiding the introduction of significant aberrations. The illumination of the sample by a partially spatially coherent source (***Deng and***

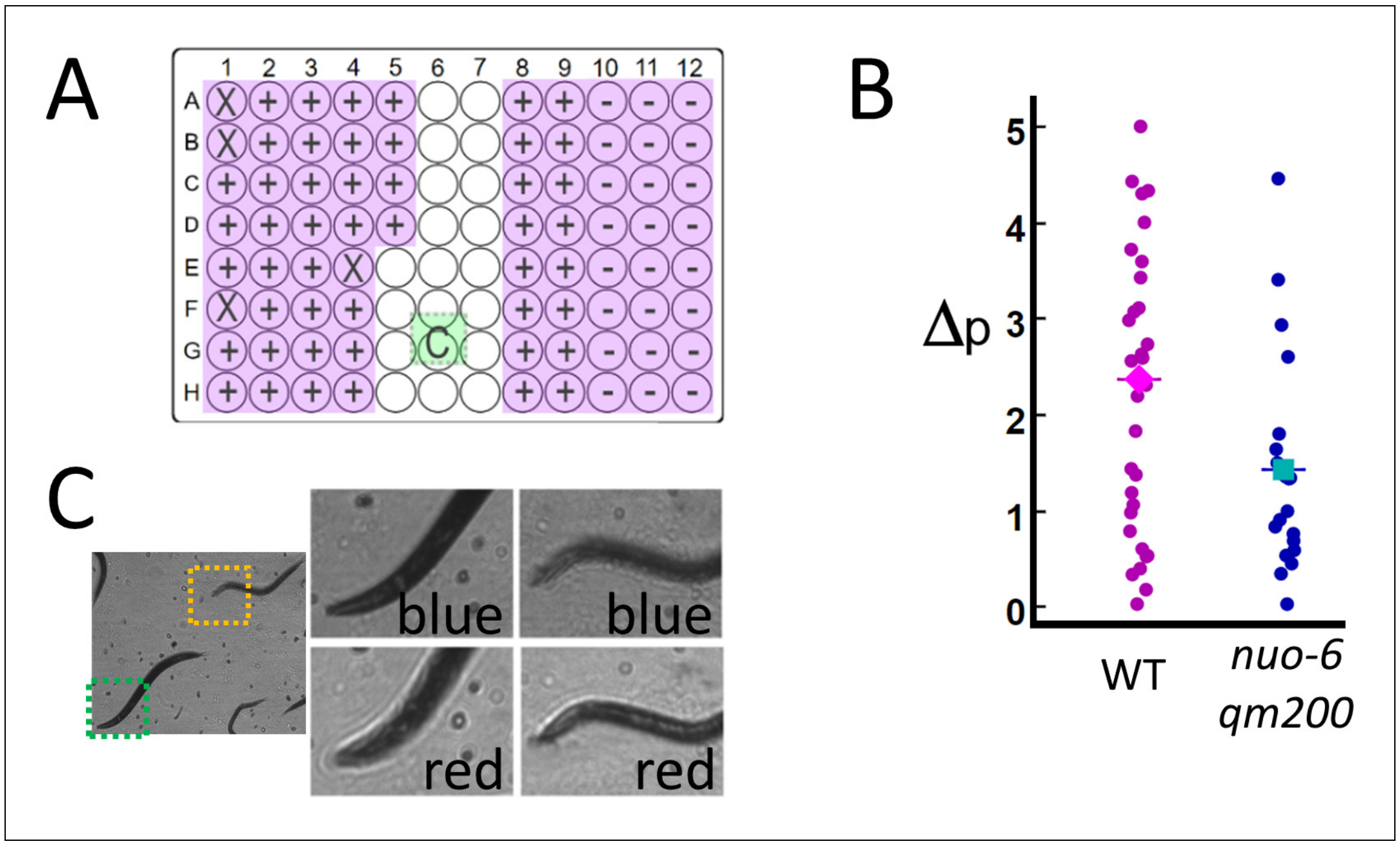


**Figure 3.** High-throughput estimates of *C. elegans* motion in liquid media. Images are captured at 120 fps, which is split over multiple wells as shown in ***Video 1***. (**A**) The position of the active detection sites (magenta) relative to the camera (green), which obscures a portion of the 96-well plate: Wells obscured by hardware are denoted by an 'X' symbol (see Materials and methods: ***Table 1***), wells with wild-type *C. elegans* (WT, '+' symbol) and mutant (*nuo-6(qm200)*, '−' symbol). (**B**) Motion analysis comparing wild type (magenta dots) to mitochondrial mutant *nuo-6(qm200)* (blue dots): wells in each row are imaged in parallel (eight wells at 15 fps per well), and net motion is estimated in each well by summing absolute differences in pixel intensities in sequential frames (see Materials and methods: Image analysis). This estimate confirms that the imaging system can detect significant differences between the two strains (averages shown by diamond and square symbols, two-tailed t-test p=0.01), which is consistent with published results (***Yang and Hekimi, 2010***). (**C**) Focal plane wavelength dependence: details from two fields of view (dashed green and orange squares) in the same image appear in or out of focus depending on whether imaged using a red or blue LED (see ***Video 2*** and ***Figure 3—figure supplement 1***).
The online version of this article includes the following figure supplement(s) for figure 3:

**Figure supplement 1.** Focal plane wavelength dependence.

***Chu, 2017***) produces greyscale images, and in our studies, it is the change in this intensity that is of interest.

Propagation-based phase contrast in our imaging system is generated when collimated light from the LED is diffracted by the sample. Light that remains in the collection cone of the objective lens is then refocused on the sensor by the parabolic reflector at an oblique angle (***Figure 1C***). As a result of this angle, the image moves through focus from one side of the detector plane to the other. The region over which the image is in focus is determined by the depth of focus of the parabolic mirror. The distance along the chief ray ($D_f$) between the image at either side of the detector is given by $D_f = D_s \sin(\theta)$, where $D_s$ is the width of the sensor and $\theta$ is the angle of the chief ray. For our system, $D_s$ is 2.4 mm, and $\theta$ is always less than 60 degrees, so $D_f$ is always less than 2 mm and the entire image therefore remains inside the Rayleigh length of the parabolic focus.

Images are subject to two transformations: (1) a stretch due to the image meeting the camera plane obliquely and (2) a small variation in magnification as a function of the separation between the optical axes of the objective lens and the parabolic reflector. These image transformations can be compensated by post-processing the captured images using equations derived from geometric optics as described below.

Light from the sample arrives at the detector plane at an incidence angle θ, which increases with lateral displacement between objective and mirror axes, *y* (***Figure 1C***). As the image itself is formed normal to the chief ray, the detector plane captures a geometric projection of the image which is stretched in the direction of *y*. The magnitude of the stretch is given by

$$S = \frac{1}{cos\left[2\, tan^{-1}\left(\frac{y}{2f_M}\right)\right]} \quad (1)$$

where $S$ is the magnitude of the stretch in one axis, $y$ is the lateral displacement, and $f_M$ is the focal length of the parabolic mirror. In addition, there is also a small variation in magnification, which is the same in both image dimensions ($y'$ parallel to displacement $y$, and $x'$ orthogonal to $y$) due to the distance between the parabolic mirror surface and the focal point ($V$) increasing as a function of $y$ (**Figure 1D**). The magnification is then given by the ratio of $V$ to the focal length of the objective lens, $f_L$. As $V(y)$ can be calculated precisely for a parabola, the magnification $M$ can be written as function of $y$, $f_L$, and mirror focal length, $f_M$:

$$M = \frac{1}{f_L}\left\{y^2 + \left(f_M - \frac{y^2}{4f_M}\right)^2\right\}^{\frac{1}{2}} \quad (2)$$

The combined magnification ($M_C = M \times S$) from global scaling and geometric projection along the $x'$ and $y'$ dimensions is shown together with measured results in **Figure 1E**.

We demonstrate the system using two popular biological models that may benefit from capturing images in parallel. Cultured cardiac monolayer preparations (**Tung and Zhang, 2006**; **Shaheen et al., 2017**) are used to study arrhythmogenesis in controlled settings and are subject to intense research due to their potential for screening compounds for personalized medicine. *Caenorhabditis elegans* are used as model organisms to study the genetics of aging and biological clocks (**Hekimi and Guarente, 2003**) and, due to highly conserved neurological pathways between mammals and invertebrates, are now used for neuroprotective compound screening (**Larsch et al., 2013**). Both model systems are ideally imaged continuously over long periods to capture dynamics (**Larsch et al., 2013**; **Kucera et al., 2000**), which is not possible in automated microscopy platforms that move samples or the optical path. The preparations were imaged using four 25 mm diameter, 100 mm focal length lenses (see Materials and methods: Configuration 1). **Figure 2A** shows recordings from four dishes imaged in parallel containing monolayer cultures of neonatal cardiac cells at 40 fps per dish. Here, motion is tracked by measuring the absolute value of intensity changes for each pixel over a six-frame window (**Burton et al., 2015**). Intensity vs time plots (**Figure 2B**) highlight different temporal dynamics in each preparation, and an activation map from one of the dishes shows conduction velocity and wave direction data (**Figure 2C**). *C. elegans* can similarly be imaged, here at 15 fps for four dishes over a period of 5 min (**Figure 2D–F**). *C. elegans* motion paths (**Figure 2D**), which are often used to quantify worm behaviour, can be extracted from each image series using open-source software packages.

We validate the potential for RAP to be used in a higher-throughput imaging application by measuring motion in *C. elegans* mitochondrial mutant *nuo-6(qm200)* (**Yang and Hekimi, 2010**), which have a slower swimming rate (frequency of thrashing) than that of the wild-type *C. elegans*. Mutant and wild-type *C. elegans* were loaded into a 96-well plate containing liquid media and imaged by using an array of 76 6 mm diameter, 72 mm focal

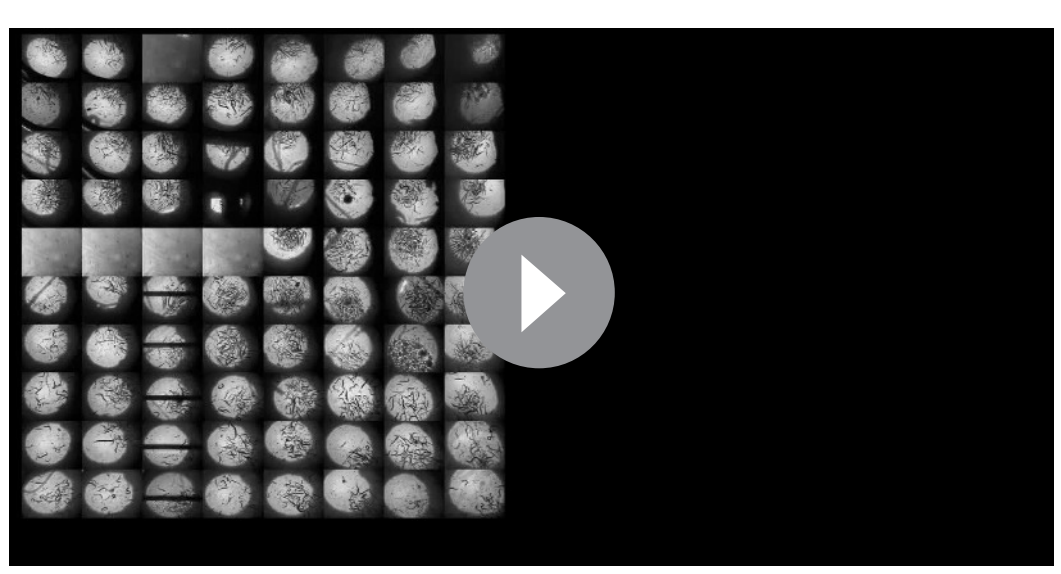

**Video 1.** RAP recordings from a 96-well plate, showing recordings at different temporal resolutions.
https://elifesciences.org/articles/56426#video1

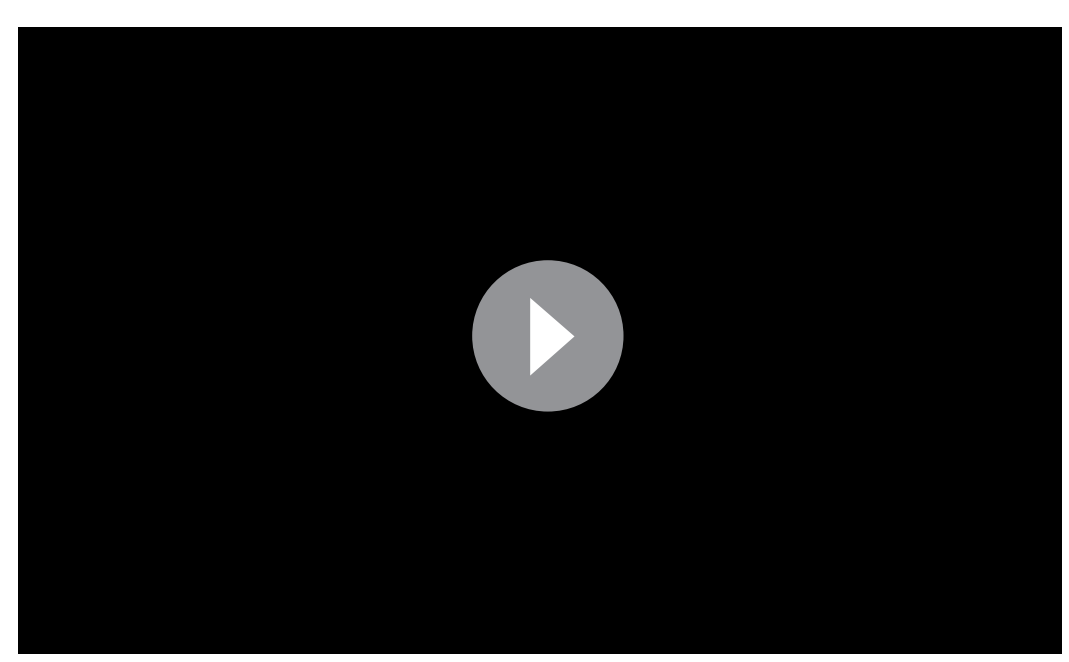

**Video 2.** RAP recordings using different colours (red and blue LEDs) focus at different planes in the sample.
https://elifesciences.org/articles/56426#video2

**Table 1.** Configuration details.
See ***Figure 1—figure supplement 1*** for additional details.

| | Configuration 1 | Configuration 2 |
|---|---|---|
| Camera | Basler acA640-750um, 750 maximum fps, with 640 × 480, 4.8 × 4.8 μm pixels | Basler acA1300-200um, 202 maximum fps, with 1280 × 1024, 4.8 × 4.8 μm pixels |
| Lenses | Edmund Optics 25 mm diameter, 100 mm focal length (NA = 0.124) | Edmund Optics 6 mm diameter, 72 mm focal length (NA = 0.04) |
| LED array | Adafruit DotStar 8 × 32 LED matrix | 2× Adafruit NeoPixel 40 LED Shields |
| Sample location | Four samples equidistant (~40 mm) from the optical axis. | Up to 76 wells in a 96-well plate (***Figure 3A***). |
| Frame rate | Images captured at 160 fps for four sample (***Figure 2A–C*** and 40.0 fps/sample) or 60 fps for four samples (***Figure 2D–F*** and 15 fps/sample). | Images captured at 120 fps for eight samples (***Figure 3*** and 15 fps/sample). Different sampling rates are shown in ***Video 1***. |
| Usage notes | Vibrations in cardiac experiments were damped by using Sorbothane isolators (Thorlabs AV5), and room light was blocked using black aluminium foil (Thorlabs BFK12). We use a 640 × 512 pixel ROI for the camera in Configuration 2 as the illumination spot is smaller than the camera FOV. Camera placement obscures 12 wells in the 96-well plate imaged in Configuration 2 (see ***Figure 3A***), and the use of two commercial 40 element LED arrays precludes imaging all wells in a 96-well plate as the LEDs are permanently mounted on a board that is too large to be tiled without leaving gaps. In addition, some wells (marked in ***Figure 3A***) were inadvertently obscured by hardware between the sample and objective lenses for the motion quantification experiment in ***Figure 3***; however, the number of imaged wells was considered to be sufficient to demonstrate the utility of the RAP system. | |

length lenses positioned above each well (see Materials and methods: Configuration 2). Instead of measuring thrashing frequency directly, motion was quantified by measuring the fraction of pixels per frame that display a change in intensity of over 25% for 100 sequential frames captured at 15 fps/well (see Materials and methods: Image processing). In this experiment, the frame rate of the camera is limited to 120 fps (see Materials and methods: Practical considerations and ***Video 1***), allowing us to image eight wells in parallel at 15 fps/well. Eighty wells (76 active and four blank wells – see ***Figure 3A***) are imaged by measuring 100 frames from each well in a row of eight wells in parallel (800 frames/row) before moving to the next row, until all 80 wells are imaged (a total of 8000 frames). The system quantified decreased activity in *nuo-6(qm200)*, which is consistent with published results (***Yang and Hekimi, 2010***; ***Figure 3B***). The time needed to perform this assay is just over 1 min (8000 frames/120 fps = 67 s).

A limitation of our current implementations of RAP is that focusing individual wells is impractical when there are more than few (i.e. four as in ***Figure 2***) active samples. For System 2 (76 wells), the objective lenses had a depth of focus of 1 mm, which is sufficient tolerance to accommodate most of the wells imaged. Small variations in lens focal length, variability in printed parts, and variations in tissue culture plates result in well-to-well variations in image quality as samples may not be perfectly in focus. While we were able to resolve *C. elegans* and measure activity in all wells, images are noticeably blurred in about half of the wells, and in some cases, some objects in a single well are better focused than others. This situation can be mitigated by changing the LED colour, as the single-element lenses used in our system show variations in focal length as a function of wavelength (***Figure 3C*** and ***Video 2***). Optical simulations using ray tracing software Configuration 2 confirm that the focal plane can be shifted by 0.981 μm by switching LED colour from red to blue (see Materials and methods). Rapid colour switching (i.e. alternating image capture between red and blue LEDs) may be used to increase data set quality at the expense of decreasing the framerate per well (as was done in ***Figure 3—figure supplement 1***) or the number of wells that can be imaged in parallel, as twice the number of images per well are required.

## Discussion

The push to develop new high-throughput screening modalities (***Abraham et al., 2004***; ***Oheim, 2007***) has resulted in several innovative approaches, ranging from the use of flatbed scanners for slowly varying preparations (***Stroustrup et al., 2013***), to wide-field methods that incorporate computational image reconstruction (***Taute et al., 2015***; ***Zheng et al., 2013***), to 'on-chip' imaging systems that plate samples directly on a sensor (***Zheng et al., 2011***; ***Göröcs and Ozcan, 2013***; ***Cui et al., 2008***; ***Greenbaum et al., 2012***; ***Greenbaum et al., 2013***). Despite these advances, methods that

**Table 2.** Comparison between conventional and RAP imaging systems.

| | Conventional microscope | RAP microscope |
|---|---|---|
| Resolution | NA = 0.025 (1×) to 0.95 (40×) | NA = 0.04 and 0.124 (1.4× and 1×) |
| Image quality | Optimal<br>(multi-element objectives correct for most aberrations) | Moderate<br>(single-element lenses used as objectives display spherical and other aberrations) |
| Modalities | Bright-field, phase contrast, DIC, fluorescence | Bright-field, multi-sample |
| Scan time* | ~8 min (no autofocus)<br>~11 min (with autofocus)† | 1 min (no focus)<br>2 min (LED colour switching) |
| Focal drift | Moderate to low (due to the use of a heavy machined platform, with further improvements afforded to autofocus systems) | Moderate to high (focal plane drift is expected due to light, 3D printed parts, but its impact can be mitigated by LED colour switching) |
| Cost | High (~$30,000 with automated x,y,z stages) | Low ($1750–$3250)‡ |
| Automation§ | Good (many automated microscopes are fully programmable) | Unknown (fully programmable, but not validated as part of a conventional high-throughput workflow) |

*Scan time is estimated for measuring the 72 unobstructed wells in a 96-well plate to allow direct comparison to the data in **Figure 3**. The estimate is based on moving serially between wells with a transit time of 0.5 s and imaging 100 frames at 15 fps. Examples from the literature vary considerably (e.g. up to one hour using 3D printed automation technologies, due to limitations in hardware communication speeds: see **Schneidereit et al., 2017**).

†We assume the autofocus algorithm takes on average 2.5 s (see **Geusebroek et al., 2000**).

‡The cost for the RAP system depends on the number of objective lenses used: Configuration 1 costs approximately $1750, while Configuration 2 (with 76 wells) costs approximately $3,250, as the cost for the cameras in both configurations are similar (~$400). Costs are in USD.

§'Automation' refers to a system's ability to be integrated into robotic workflows. Conventional automated microscopes are core components of high-throughput screening platforms with sample and drug delivery capabilities. While our system is in principle compatible with these technologies (e.g. by leveraging existing open-source software, see **Booth et al., 2018**), it has not been tested in these settings.

accommodate a biologists' typical workflow – for example comparing multiple experimental samples plated in different petri dishes – depend on automation of conventional microscopes.

Automated microscopes excel at applications where data can be acquired from samples sequentially as a single high-numerical-aperture (NA) objective is used. While a RAP system could be built using high-NA, high-magnification optics, this likely would require that each objective lens is independently actuatable in order to achieve focus which poses practical limits on the number of imaged wells. RAP systems can be used to speed up conventional imaging tasks in low-magnification settings by capturing data from different samples in parallel (as was done in **Figure 3**). However, here the speed increase afforded by RAP must be weighed against the many benefits of using a mature technology such as the automated widefield microscope (see **Table 2** for a comparison between these systems). RAP systems are better suited for dynamic experiments where multiple continuous long-duration recordings are the primary requirement. For example, rhythms in cultured cardiac tissue evolve over hours (**Kim et al., 2009**) or even days (**Woo et al., 2008**; **Burridge et al., 2016**), but display fast transitions between states (e.g. initiation or termination of re-entry **Bub et al., 1998**), necessitating continuous measurement. In these experiments, moving between samples would result in missed data. RAP overcomes these constraints by reducing transit times between samples to less than a millisecond without the use of automation or relying on a widefield imaging approach, while allowing for an optimized field of view.

# Materials and methods

## Sample preparation and imaging

Wild-type *C. elegans* were maintained in standard 35 mm petri dishes in 5–8 mm of agar seeded with *E. coli* for the data in **Figure 2**. For **Figure 3**, the mitochondrial mutant *nuo-6(qm200)* (**Yang and Hekimi, 2010**) was used along with wild-type *C. elegans*. Here, *C. elegans* were transferred to 96-well plates by washing adults off NGM plates in M9 buffer, washed once to remove *E. coli*, and resuspended in fresh M9 buffer. Fifty microlitres of this worm suspension was loaded into a 96-well, flat-bottom assay plate (Corning, Costar), excluding half of row five and all wells in rows 6 and 7 as shown in **Figure 3A**, as these wells were either obscured by sensor hardware or not illuminated by the two 40-element LED arrays (see Configuration 2 in **Table 1** below). Wells are filled with

M9 buffer and covered with a glass coverslip to reduce refraction artefacts at the meniscus interface at well borders. For additional details, see **Hekimi and Guarente, 2003**. All experiments involving *C. elegans* were imaged at room temperature. Cardiac monolayer cultures were prepared from ventricular cells isolated from 7 day old chick embryos: cells were plated within 1 cm glass rings in 35 mm petri dishes as described in **Bub et al., 1998**. Cardiac monolayers were imaged in a stage top incubator (Okolabs) at 36˚C and at 5% $CO_2$ in maintenance media.

## Optical setup

A parabolic reflector (220 mm diameter, 100 mm focal length, Edmund Optics) was mounted 300 mm above a breadboard. The camera sensor and electronics (acA640-750um for data collection in **Figure 2**, acA1300-200um for data collection in **Figure 3**, Basler AG) were mounted in a PLA (polylactic acid) housing without a c-mount thread to allow image formation from light at oblique angles and positioned at the focal point of the parabola. Biological samples were positioned 50 mm above a LED array (DotStar 8 × 32 LED matrix for **Figure 2** or two NeoPixel 40 LED Shields for **Figure 3**, Adafruit Industries). Plano-convex lenses (25 mm diameter, 100 mm focal length for **Figure 2**, 6 mm diameter, 72 mm focal length for **Figure 3**, Edmund Optics) were positioned at their focal lengths above each sample. Axial alignment tolerances were set by the depth of field ($DOF$) of the lenses, calculated to be 0.9 mm using the approximation: $DOF = (2u^2Nc)/f^2$ where the subject distance, $u$=$f$, the f-number, $N$=12, and the circle of confusion c, was set to be twice the lateral resolution (18 µm). The LED array was controlled by an ATmega328P microcontroller (Arduino Uno, Arduino.cc) using the FastLED 3.2 open-source library and custom code (**Source code 1** and **2**, in conjunction with free Basler Pylon Viewer software) to synchronize the camera with each LED via a TTL trigger pulse. Custom parts were printed with a Prusa I3 MK2S printer; STL files with an image of the setup showing their use is provided in 'stl_files.zip'. Table 1 summarizes features of the two systems.

## Image processing

We find that image brightness drops with increased objective lateral distance and that images are subject to aberrations at the edges. To offset these effects, captured images shown in **Figures 2** and **3** are cropped (480 × 480 pixels for Configuration 1, and 640 × 512 for Configuration 2) and rescaled (so that maximum and minimum pixel intensity values fall between 0 and 255). Dye-free visualization of cardiac activity (**Figure 2B**) is carried out by applying a running background subtraction followed by an absolute value operation on each pixel:

$$P_t(i,j) = |\, P_t(i,j) - P_{t-n}(i,j)|$$

where $P_t(i,j)$ is the value of pixel p at location *i,j* at time *t* and $P_{t-n}(i,j)$ is the value of the same pixel at an earlier frame (typically six frames apart: see **Burton et al., 2015**) for details on this technique). Intensity vs time plots of averaged pixels in a 20 × 20 pixel region of interest show double spikes corresponding to contraction followed by relaxation (**Figure 2B**). Activation maps (**Figure 2C**) are generated as previously described (**Burton et al., 2015**). Motion (**Figure 3B**) is quantified by finding the magnitude of the intensity change between co-localized pixels in sequential images, counting the number of pixels where the magnitude of the change is over 65 intensity units (25% of the intensity range of the image), and dividing the total by the number of analysed frames. We note that while this algorithm yields results that are consistent with published manual measurements of thrashing frequency (see figure 2j in **Yang and Hekimi, 2010**), there is no direct correspondence between this metric and specific behaviours (head movement, posture changes, etc.). However, the documented difference in the activity of the two strains we use would predict the difference in the metric that we observe and can be used as a validation of the imaging method to track movement over time.

## Practical considerations

The camera used in **Figure 2** was chosen for its high frame rate as we were interested in imaging cardiac activity, which in our experience requires 40 fps acquisition speeds. The small field of view imposed by the sensor (640 × 480 pixels at 4.8 microns per pixel giving a 3 × 2.3 mm FOV for the 1× imaging scheme used in **Figure 2**) was considered reasonable as the field imaged by the 25 mm lens was larger than the sensor, ensuring that the sensor will always capture useful data. In contrast, the system used in **Figure 3** used smaller 6 mm lenses, and a relatively small 4 mm diameter spot

was projected on the sensor. Small changes in lens angle and position (which proved to be hard to control using our consumer grade desktop 3D printer) result in up to a millimetre well-to-well variation for position of the image on the sensor. We therefore opted to use a higher resolution camera with a larger sensor to ensure that the image would reliably fall on the sensor. While this choice lowers the number of frames that can be continuously saved to disk, we considered this to be an acceptable trade-off as the frame rate needed to image *C. elegans* motion is relatively modest. Future designs will use precision (e.g.CNC (computer numerical control) machined) lens holders that would reduce these variations by an order of magnitude.

The imaging scheme captures data at a maximum rate that depends on the camera as well as the system's ability to save data continuously to disk. Our system's hard drive is capable of continuously saving to disk at 150 MB/second. The camera used in Configuration 2 has a resolution of 1280 × 1024 pixels, which generates 1.25-MB images: the 150 MB/second limit therefore imposes a sustained base frame rate of 120 fps (150 MB/second/1.25MB = 120 fps). *C. elegans* motion can be adequately quantified when imaging at 15 fps, allowing us to image eight wells (120 fps/15 fps) in parallel. A faster hard drive (e.g. an SSD) or RAID array would significantly increase throughput.

We note that RAP has been validated in low-magnification, bright-field settings that have relaxed constraints relative to microscopy applications that may require high magnification with optimized resolution and high light throughput (e.g. fluorescence microscopy). Rather, our designs aim to maximize the number of independent samples that can be imaged in parallel. We therefore opt to use inexpensive components and minimize the device's footprint, allowing us to either increase the number of samples captured by a single system or alternatively – as large parabolic reflectors may not be practical in a lab setting – duplicate the system to increase total capacity.

The use of low-magnification optics in our current implementation is not a defining property of RAP, as higher NA, high-magnification optics could be used. In the same way that the objective lens is not limited by the tube lens in a conventional microscope, the choice of the objective lenses in the RAP microscope is not limited by the parabolic mirror. The NA (and resolving power) of the implementations described above to demonstrate RAP microscopy are consistent with other low-magnification systems. Conventional bright-field 1× microscope objective lenses have NAs close to that of Configuration 2 (e.g. the Zeiss 1× Plan Neofluar commercial objective has an NA of 0.025, and the Thorlabs TL1X-SAP 1× objective has an NA of 0.03), and research stereo macroscopes have NAs close to that of Configuration 1 (e.g. the NA is 0.11 for an Olympus SZX10 at 1×), but can be higher in specialized macroscope systems. As is the case with conventional microscope designs, a high-magnification RAP system would likely require a mechanism for finely adjusting objective heights to keep each sample in focus, as the depth of field of the objective lenses would be reduced. While the resolution of a RAP system is similar to conventional microscopes, RAP systems differ from conventional microscopes in several respects. ***Table 2*** summarizes some key differences between a conventional automated widefield imaging microscope and the two RAP systems implemented in this publication. We note that higher performance RAP systems (e.g. faster disks, a faster camera, corrected optics) would display improved performance.

## Optical model validation

To validate the optical model of the imaging system (***Equations 1 and 2***), an opaque grid with a 200 µm pitch (#58607, Edmund Optics) was used as a test sample. Images of grid sample were captured using an objective lens with its optic axis separated from that of the mirror by distances shown in ***Figure 1E***. Rescaling the images by the factor given in ***Equation 1*** recovers the image of the square grid.

## Optical resolution comparison

To compare the performance of RAP (Configuration 1) to a conventional on-axis imaging system, the parabolic mirror was replaced by a plano-convex lens with the same 100 mm focal length and aligned co-axially with the objective lens and sample. A qualitative comparison of images of a US Air Force chart showed that image resolution degradation in the RAP system, caused by off-axis aberrations in the parabolic mirror, is relatively modest for small (<40 mm) axial distances (***Figure 1—figure supplement 2***).

**Table 3.** Comparison of image quality (intensity contrast and estimated lateral width of the point spread function) for varying distances from the optic axis.

| Off-axis distance (mm) | Contrast at 25 lp/mm | Estimated FWHM (µm) |
|---|---|---|
| 22.16 | 4.50 | 14.9 |
| 29.96 | 6.52 | 13.4 |
| 38.48 | 6.06 | 13.7 |
| 45.04 | 3.62 | 16.0 |
| 53.90 | 3.27 | 16.6 |
| 60.46 | 2.101 | 20.3 |
| 66.84 | 1.88 | 21.6 |

In addition, images of an optically opaque grid were captured on the Configuration 2 system for a variety of off-axis distances. The intensity contrast (the ratio of the darkest region in the gridline to the intensity in the adjacent transmissive region) was used to infer the lateral extent of the optical point spread function (PSF) by comparison to a computational model. The model calculated the anticipated contrast as a function of PSF width (PSF FWHM, see below) using a simple convolution. As the original width of the gridline was known (20 µm, equivalent to 25 line pairs/mm), this relationship could then be used to estimate the lateral PSF width for a given intensity contrast (***Table 3***). The theoretical lateral resolution of a 6 mm diameter, 72 mm focal length lens was calculated to be: $PSF(XY) = 0.6 * \lambda/NA = 9.1\ \mu m$ when using the centre emission wavelength of 622.5 nm from the Adafruit Neopixel red LEDs. Estimated lateral PSF widths varied from 13.4 to 21.6 microns over the full range of axial distances used in the 96-well experiment, with performance falling as a function of axial distance.

## Optical simulations

The chromatic focal shift observed in the experiments was confirmed using optical simulations (Zemax OpticStudio 18.1). The shift in the back focal plane, solved for marginal rays at a particular wavelength, was calculated. For the plano-convex lens used in Configuration 2 (Edmund Optics #45–696), this focal shift was found to be 981 µm when switching from a red (622 nm) to blue (469 nm) LED.


# Acknowledgements

We thank RS Branicky and S Hekimi for the *C. elegans* preparation, A Caldwell for sample preparation, and C Sprigings for programming assistance.


# Additional information


### Funding

| Funder | Grant reference number | Author |
|---|---|---|
| National Science and Engineering Research Council of Canada | RGPIN-2018-05346 | Gil Bub |
| National Science and Engineering Research Council of Canada | RGPIN-2016-05962 | Simon Thibault |
| Heart and Stroke Foundation of Canada | HSFC G-18-0022123 | Gil Bub |

The funders had no role in study design, data collection and interpretation, or the decision to submit the work for publication.

### Author contributions

Mishal Ashraf, Investigation, Writing - original draft; Sharika Mohanan, Software, Formal analysis, Investigation; Byu Ri Sim, Kiamehr Rahemipour, Investigation; Anthony Tam, Software, Investigation; Denis Brousseau, Simon Thibault, Investigation, Writing - review and editing; Alexander D Corbett, Supervision, Investigation, Methodology, Writing - review and editing; Gil Bub, Conceptualization, Resources, Software, Supervision, Investigation, Methodology, Writing - original draft, Writing - review and editing

### Author ORCIDs

Alexander D Corbett https://orcid.org/0000-0003-1645-5475
Gil Bub https://orcid.org/0000-0002-5304-0036

### Decision letter and Author response

Decision letter https://doi.org/10.7554/eLife.56426.sa1
Author response https://doi.org/10.7554/eLife.56426.sa2

## Additional files

### Supplementary files

- Source code 1. Arduino code for controlling the camera and LED array for Configuration 1.
- Source code 2. Python code for sorting images for each sample into unique directories.
- Supplementary file 1. STL files and instructions for assembling RAP Configurations 1 and 2.
- Transparent reporting form

### Data availability

All data generated during this study are included in the manuscript and supporting files.

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
