## [Decision Letter]

**Acceptance summary:**

The clarifications that you have made now allow readers to judge for themselves the utility of this novel imaging modality. While the current system falls short of providing truly continuous high-speed (i.e. > 10 fps) imaging of 80-96 wells, it clearly has great potential for multiple applications.

**Decision letter after peer review:**

Thank you for submitting your article "Solid state high throughput screening microscopy" for consideration by *eLife*. Your article has been reviewed by three peer reviewers, and the evaluation has been overseen by a Reviewing Editor and Didier Stainier as the Senior Editor. The following individual involved in review of your submission has agreed to reveal their identity: Didier Marguet (Reviewer #1).

The reviewers have discussed the reviews with one another and the Reviewing Editor has drafted this decision to help you prepare a revised submission.

As the editors have judged that your manuscript is of interest, but as described below that additional experiments are required before it is published, we would like to draw your attention to changes in our revision policy that we have made in response to COVID-19 (https://elifesciences.org/articles/57162). First, because many researchers have temporarily lost access to the labs, we will give authors as much time as they need to submit revised manuscripts. We are also offering, if you choose, to post the manuscript to bioRxiv (if it is not already there) along with this decision letter and a formal designation that the manuscript is “in revision at *eLifeeLife*”. Please let us know if you would like to pursue this option.

The reviewers all recognised the originality of your solution to perform high throughput imaging without moving parts. They do have some serious reservations, primarily regarding the evaluation of the quality and utility of the technique and in addition to the other points raised, consider it essential that you address the following:

– The standard topics for any new microscope paper: "objective" numerical aperture, image resolution, optical aberration, and camera sensor size, together with the specific aspects related to this technique, including dependence on homogeneous illumination, and sensitivity to maintenance of F2 distance.

– A substantial expansion of the scope of the data presented, to provide readers with sufficient evidence with which to evaluate the quality of the technique, including proof of principal with a 96-well plate assay.

– A direct quantitative comparison with existing HTS imaging solutions.

Reviewer #1:

Ashraf and colleagues describe an approach to perform high throughput screening imaging without moving parts. The setup is original and offers to experimentalists the flexibility to record quasi-simultaneously stacks of images of multiple samples at the full field of resolution of the camera. The optical aberration inherent to the use of a parabolic mirror are mostly overcome by collimating light from the objective lens. The images require a post-processing in two steps for taking into account the image stretching on the detector and the variation in magnification due to the variation of the distance between the mirror and the image. Two applications illustrate the potential of the solid-state HTS.

To my opinion, the following points need to be clarified:

– How homogeneous is the field of illumination with a single LED? Especially for a large field of illumination, a non-homogeneous illumination would compromise the quantifications.

– The accuracy of this ssHTS is related to the robustness at keeping the distance F2 constant between samples. In other words, how sensitive is the image acquisition to the potential variation in the F2 distance between samples as well as within a single large field of view?

– The magnification Mc must be explained.

– Is the post-processing compensation applied only in the y-direction?

Assuming that such publication aims to disseminate the use of an ssHTS setup to a wide scientific community, I find the description of the setup as well as the applied image post-processing rather succinct, even with the 3D printing and source codes information.

Reviewer #2:

Astronomers have spent centuries learning how to image the night sky with limited sensor hardware. Ashraf et al. present an ingenious adaptation of a technology developed for telescopes-parabolic reflectors-for imaging biological samples. In principle, the approach seems like it could be incredibly useful across a wide range of applications where multiple samples must be imaged in tandem. By placing multiple samples under a single parabolic reflector, multiplexing of samples and imaging hardware can be accomplished without sample-handling robots or moving cameras. The authors highlight two applications: cardiac cells in culture and free-moving nematodes.

The authors explain the theory behind their technique in a clear and convincing way. However, the biggest challenge in most imaging projects is making the theory work in practice. In its current form, the manuscript falls far short of demonstrating the practical usefulness of parabolic mirrors for imaging biological samples. The authors include only a small amount of image data-for the nematode work, this consists of eight images collected from two plate regions. Data of this scope cannot provide readers or reviewers with sufficient evidence with which to evaluate the quality of the technique.

1) The images shown-are they typical or are they the best possible images that can be collected from the device? The authors do not provide any quantitative evaluation of the quality of their images, in absolute terms or relative to existing methods, with which to understand the practical performance of parabolic mirrors. The authors should estimate the spatial resolution and dynamic range that can be obtained in practice with the devices, and evaluate how such image quality metrics vary across the entire field of view. Does performance degrade towards the edge of the mirror? Does performance degrade over time, as devices become de-calibrated with use?

2) The manuscript is additionally weakened by the absence of a non-trivial measurement made with the device. Pilot experiments are included, demonstrating that images can be collected. However, no evidence is provided to show that these images can be used to compare samples and draw biological conclusions from them. A more convincing proof-of-principle would involve the measurement of some non-trivial biological difference between samples measured with the device, either confirming previous work or discovering something new.

3) The authors highlight the comparative simplicity of their method: it eliminates the need for motorized samples or cameras. However, this simplicity must come at some: for example a substantially increased use of space or perhaps an increase in delicate calibration required, or equipment price. If a 0.25 meter mirror is required to measure four *C. elegans* plates, how large a mirror would be required to measure 16 plates-the number that can typically be measured using a flatbed scanner? The authors could also expand greatly on other practical issues: for example, is a dedicated imaging table required to align mirrors and samples? Readers would benefit from a clearer evaluation of the practical trade-offs in deploying parabolic mirrors in a laboratory setting relative to other imaging approaches.

Reviewer #3:

The authors present a cool new idea: using a large parabolic reflector in combination with a macroscopic lens array and rapidly modulated LED array to enable fast image multiplexing between spatially separated samples. I believe that there may be interesting applications that would benefit from this capability, although the authors have not clearly demonstrated one. The paper is short, and light on discussion, details, and data.

1) The manuscript does not discuss several standard, key topics for any new microscope paper: "objective" numerical aperture, image resolution, optical aberration (other than distortion, which is discussed), and camera sensor size.

2) Why was an array of low-performance singlet lenses used? With that selection, the image quality cannot be good. Can the system not be paired with an array of objectives or higher performance multielement lenses?

3) Fluorescence imaging is not discussed or demonstrated but would obviously increase the impact of the microscope. At least some discussion would be helpful.

4) Actual HTS applications are almost always implemented in microtiter plates (e.g. a 96-well plate) to reduce reagent costs and enable automated pipetting, etc. I do not believe anyone would implement HTS in thousands of petri dishes. The paper would be strengthened substantially by a demonstration of simultaneous recording from all (or a large subset) of the wells in a 96-well plate. It's not clear whether this is possible due to the blind spot in the center of the parabolic mirror's field of view that is blocked by the camera.

5) One of the primary motivations for this approach is given in the first paragraph as: "wide-field imaging systems [which capture multiple samples in one frame] have poor light collection efficiency and resolution compared to systems that image a single sample at a given time point." With a f = 100 mm singlet lens, the light collection efficiency of the demonstrated microscope is also low (estimated NA = 0.12) and the resolution is unimpressive with the high-aberration lens and 1x magnification. They demonstrated only trans-illumination applications (e.g. phase contrast), where light collection efficiency is not important. I believe a fancy photography lens mounted directly on a many-megapixel camera set to image all or part of a microtiter plate could likely outperform their system in throughput and simplicity, at least for the demonstrated applications of cardiomyocytes and *C. elegans*.

[Editors' note: further revisions were suggested prior to acceptance, as described below.]

Thank you for resubmitting your work entitled "Solid state high throughput screening microscopy" for further consideration by *eLife*. Your revised article has been evaluated by Didier Stainier (Senior Editor) and a Reviewing Editor following review and discussion by the three original reviewers.

Overall, the reviewers recognised that this setup could be useful for readers looking for an inexpensive bright-field imaging setup for multi-well imaging without fluorescence. They agree that you have provided substantial additional data and analysis that support your claim that parabolic reflectors can be useful for studying many samples in a parallel. The new images and videos of a 96-well plate were judged compelling. In particular, the prospect of focusing samples simply by adjusting the wavelength of illumination was thought an important step towards the goal of designing a "solid-state" imaging apparatus with no moving parts.

Nevertheless, although the reviewers were satisfied that you had addressed most or all of the material points, they had considerable reservations about the way in which these improvements were presented and could not support publication of the manuscript in its present form. Indeed, there are numerous lacunae and parts where the writing is not at all clear and leads to confusion about how the system functions and what its limits are. Please find below a summary of their most important comments and a series of points made by individual reviewers, all of which would need to be addressed in a revised manuscript. If this would require a further round of experimental work, then I am afraid that we will not be able to consider your work for publication.

The description of the 96-well plate data was considered both terse and vague, leaving unclear several aspects of experimental design and interpretation.

– If no samples were loaded into columns 6 and 7 of the 96 well plate because of the use of 40 LED arrays, this should be stated explicitly.

– What was the exact reason for not imaging in column 4 of row E or row 1F.

These discrepancies between the theoretically predicted function of the device and its practical performance must be clarified.

If these issues do not reflect technical limitation of the device, you would need to demonstrate that these columns/wells can be imaged just like the others (i.e. this is a criterion for rejection).

The details about acquisition are so poorly described that one reviewer wrote, "why not leverage those capabilities to scan 33 wells in parallel at 15 Hz rather than one well at a time at 15 Hz?". This illustrates how you have failed to convey clearly that the system captures data from multiple wells in parallel at 120-500 fps. One video does show how 120 fps can be divided up across 80 wells, and it is illustrated in Figure 1, but these details need to be explicitly stated in the text. In Figure 2, a faster (500 fps) camera of lower resolution is used. As well as making all acquisition details clearer, you will need to provide an explicit discussion of camera choice, and any trade-off between image resolution and speed. Additionally, you need to address another technical limitation and trade-off, namely rates of acquisition and data transfer so that the possibility (and cost) of implementation in a HTS setting (see below) is clear.

The center of the optical system is intrinsically blind since space is required to position the detector. This point is implicit and must be documented as a function of the magnification.

The microscope resolution in the 15 – 20 µm range is poor relative to the sub-micron resolution of a traditional microscope. It is probably not good enough, for example, to tell individual mammalian cells apart in a confluent monolayer. This will limit the range of potential applications. Thus the spatial resolution needs to be stated in the Abstract or Introduction, not buried deep in the Materials and methods. Further, you will need to include a detailed comparison with a standard commercial widefield microscope with a scanning stage (resolution, imaging modalities, scan time, defocusing over time, cost, integration into robotic workflows). If you wish to claim HTS capacity, the comparison should also include a dedicated commercial HCS/HTS system, and the many other features needed for HTS (e.g. see https://www.ncbi.nlm.nih.gov/books/NBK558077/).

Alternatively, in the absence of easy incorporation of the system in an automated setting, at a time when HTS can mean >50,000 tests/day, "High Throughput" should be removed from the title ("multi-sample" or "multi-well" would be better), and any suggestion in the text that your system is HTS-compatible seriously toned down. Equally, given the very different uses in optics or electronics of the term "solid-state", you should avoid it in the title, replacing it, for example by, "with no moving parts".

There was also a general consensus that your design is not a Newtonian telescope, which has two mirrors instead of a single mirror as in this design. The reviewers recommend changing "novel Newtonian telescope design" to "large on-axis parabolic mirror design", "parabolic reflector", or something similar that is clearer and more accurate. Including a phrase like "inspired by a Newtonian telescope" would be acceptable.

Further points made by individual reviewers:

1) The authors compare wild-type *C. elegans* to nuo-6 mutants. The authors are vague and qualitative in their descriptions of movement. Nuo-6 mutants are predicted to "move less frequently" than wildtype. This is confusing, as *C. elegans* generally exhibit some degree of continuous movement as long as they remain alive, involving body postural changes, head movements, or pharyngeal pumping. Are the authors referring to the frequency of a particular type of movement? For the purposes of this paper, the authors probably do not need to alter their imaging pipeline, but they should be substantially more specific about which behavior their method is measuring.

2) Many nematode behaviors change in response to stimulation with light, physical stimulus, or immersion in liquid. Other behaviors are suppressed by long periods spent immersed in un-mixed liquids. It remains difficult to interpret the authors' results without additional information describing how the light and culturing conditions they are use influences nematode behavior and how this influences their results. In particular, the behavioral difference observed between day 1, 2 and 3 could be expected as a technical artifact (i.e., in the absence of any underlying aging process) if nematodes remained in the same wells for multiple days.

3) The authors observe a difference in activity between nuo-6 and wild-type animals, and also between young animals and old animals. However, discussion of this is surprisingly qualitative given the quantitative thinking found elsewhere in the paper. Are the observed differences in movement approximately the same magnitude as what would be expected given previous results? Why is a significant difference between the two strains observed only on day one and three, but not day two?

4) The Figure 1 caption uses fM and fL while Figure 1 uses F1 and F2. Please make consistent.

5) Equation 2 is not fully displayed.

6) Introduction: Please give some concrete examples of experiments that require continuous long-term recording where low-resolution brightfield imaging would be the appropriate readout modality.

7) Introduction: The phrase "high resolution" is misleading, as the 15-22 µm resolution of this microscope would be considered very low resolution by most microscopists. Please insert the actual resolution here.

8) Results: I would not call this a high light collection efficiency design, as most standard microscopes have higher efficiency. Light collection efficiency is not very important here, so please change the language to be less contentious.

9) Results: Calling an LED source spatially coherent is really straining the definition. Please use different language.

10) Materials and methods: Something is wrong or confusing about the depth of focus discussion. Please cite a source for the equations and clearly define all variables. If u = f as you indicate in the text, then DOF=2c≠0.9 mm, which was stated in the text. The f-number does not appear in the equation you have, but the discussion seems to indicate that it is important (as would be expected).

11) Figure 2 legend: should be "(blue trace in B)"

12) Figure 3 legend: duplicated text "C) Focal plane…."

13) The authors limit their discussion of statistical analysis of animal movement to the legend of Figure three. This analysis would seem more natural to include either in the main text or in a dedicated statistical methods section

14) Provide more precise references to allow others to set up an ssHTS system; see for example the references for LEDs.

[Editors' note: further revisions were suggested prior to acceptance, as described below.]

Thank you for resubmitting your work entitled "Random Access Parallel Microscopy" for further consideration by *eLife*. Your revised article has been evaluated by Didier Stainier (Senior Editor) and a Reviewing Editor.

The manuscript has been improved but there are some remaining issues that need to be addressed before acceptance, as outlined below:

The authors stress that one of the principal interests of the system is the capacity for rapid and continuous imaging. They write, "captures data 15 fps/well by measuring groups of eight wells in parallel". Then they write, "As the system captures data from multiple wells in quick succession at a rate of 120 fps, the time needed to acquire 100 frames for each of the 76 wells for this assay is just over one minute". They need to be more explicit. When they are capturing data from 76 wells, then are they imaging each well at ca. 1.5 fps? As it stands, a reader might understand that they are switching between groups of eight wells, imaging one group at 15 fps/well, then moving to the next group after capturing 100 frames (6.7 seconds). If this were the case, then they would return to image the first group after a minute, so their system would not be continuous. This clearly requires clarification.

---

## [Author Response]

The reviewers all recognised the originality of your solution to perform high throughput imaging without moving parts. They do have some serious reservations, primarily regarding the evaluation of the quality and utility of the technique and in addition to the other points raised, consider it essential that you address the following:– The standard topics for any new microscope paper: "objective" numerical aperture, image resolution, optical aberration, and camera sensor size, together with the specific aspects related to this technique, including dependence on homogeneous illumination, and sensitivity to maintenance of F2 distance.

We have addressed these issues by making several improvements to the paper. First, the system is now better described, with figure supplements (Figure 1—figure supplement 1) and tables (Materials and methods: Table 1) The resolution of the system has been quantified, both qualitatively (Figure 1—figure supplement 2) and quantitatively (Materials and methods: Table 2). In addition, we found that we can move the F2 distance dynamically by almost a millimeter by switching LED wavelength, which greatly simplifies focal plane issues (Figure 3C, Figure 3—figure supplement 1, and Video 2). The text also now includes details relating to specific issues raised by the reviewers (discussed below).

– A substantial expansion of the scope of the data presented, to provide readers with sufficient evidence with which to evaluate the quality of the technique, including proof of principal with a 96-well plate assay.

We have implemented a multiwell imaging system based on imaging most of the wells in a 96-well plate (See Video 1) and used this to perform a proof of principle study on C-elegans mutants with reduced activity (Figure 3A and B).

– A direct quantitative comparison with existing HTS imaging solutions.

As there are many HTS systems, we decided to address this issue by comparing the images collected using our platform to those collected using a standard on-axis optical path, which most microscopes and HTS systems use (Figure 1—figure supplement 2, and new section “Image quality quantification” in Materials and methods). We compare the performance both to a theoretical calculated maximum as well as to a setup that uses the same lenses but in a more standard configuration.

Reviewer #1:Ashraf and colleagues describe an approach to perform high throughput screening imaging without moving parts. The setup is original and offers to experimentalists the flexibility to record quasi-simultaneously stacks of images of multiple samples at the full field of resolution of the camera. The optical aberration inherent to the use of a parabolic mirror are mostly overcome by collimating light from the objective lens. The images require a post-processing in two steps for taking into account the image stretching on the detector and the variation in magnification due to the variation of the distance between the mirror and the image. Two applications illustrate the potential of the solid-state HTS.To my opinion, the following points need to be clarified:– How homogeneous is the field of illumination with a single LED? Especially for a large field of illumination, a non-homogeneous illumination would compromise the quantifications.

As the reviewer correctly points out, the illumination field is not particularly homogeneous as we do not use any collimating optics above the LED. We find, however, that field flatness is not essential for the biological studies we typically conduct, as our examples involve looking at the differences between images. The intensity of a pixel in any one frame is effectively normalized in these measurements (either by background subtraction or rescaling based on the maximum and minimum values of that pixel’s intensity over the duration of the recording). This has been made clearer in the text: “The illumination of the sample by a spatially coherent source produces grey scale images, and in our studies, it is the change in this intensity that is of interest.”

We should note that if field flatness was indeed needed, collimating optics could be added over each LED but this would increase system cost.

– The accuracy of this ssHTS is related to the robustness at keeping the distance F2 constant between samples. In other words, how sensitive is the image acquisition to the potential variation in the F2 distance between samples as well as within a single large field of view?

Sensitivity to variations in the sample-objective separations are determined by the Rayleigh length of the objective lenses. As long as the samples remain within the Rayleigh length (approx. 1 mm for a 100 mm focal length lens), a sharp image of the sample will be formed. In addition, we observe a strong dependence of focal plane location on LED wavelength which we confirmed using Zemax optical simulations: red and blue LED illumination result in images from planes that are roughly 1mm apart. As alignment is maintained to <2 mm by the Thorlabs cage system, we find that most of wells are sufficiently in focus to resolve samples. Figure 3C now has an example of an image that has samples (*C. elegans*) at slightly different planes and shows how this can be corrected by changing LED colour. Video 2 gives an example where LED colour is switched rapidly to obtain pairs of images, allowing selection of the best image for analysis. Figure 3—figure supplement 1 shows all wells from a single row of a 96 well plate, which can be used to assess sensitivity to small variations in F2 distance (i.e. by wells captured at a single LED colour).

– The magnification Mc must be explained.

We thank the reviewer for catching this. Mc is now defined on : “The combined magnification (*M_C_*=*M***S*)”.

– Is the post-processing compensation applied only in the y-direction?Assuming that such publication aims to disseminate the use of an ssHTS setup to a wide scientific community, I find the description of the setup as well as the applied image post-processing rather succinct, even with the 3D printing and source codes information.

We agree that our original submission was missing needed details. We’ve expanded the paper, including images of the setup (Figure 1—figure supplement 1), and a table in the Materials and methods section that give additional details (Table 1, Materials and methods). Post-processing compensation is applied in both x and y directions, as described in the caption of Figure 1. The small amount of residual geometric correction has been applied in both axes. The distortion correction algorithm is a generic algorithm that does not take into account the specific geometry of the mirror.

Reviewer #2:Astronomers have spent centuries learning how to image the night sky with limited sensor hardware. Ashraf et al. present an ingenious adaptation of a technology developed for telescopes-parabolic reflectors-for imaging biological samples. In principle, the approach seems like it could be incredibly useful across a wide range of applications where multiple samples must be imaged in tandem. By placing multiple samples under a single parabolic reflector, multiplexing of samples and imaging hardware can be accomplished without sample-handling robots or moving cameras. The authors highlight two applications: cardiac cells in culture and free-moving nematodes.The authors explain the theory behind their technique in a clear and convincing way. However, the biggest challenge in most imaging projects is making the theory work in practice. In its current form, the manuscript falls far short of demonstrating the practical usefulness of parabolic mirrors for imaging biological samples. The authors include only a small amount of image data-for the nematode work, this consists of eight images collected from two plate regions. Data of this scope cannot provide readers or reviewers with sufficient evidence with which to evaluate the quality of the technique.1) The images shown-are they typical or are they the best possible images that can be collected from the device? The authors do not provide any quantitative evaluation of the quality of their images, in absolute terms or relative to existing methods, with which to understand the practical performance of parabolic mirrors. The authors should estimate the spatial resolution and dynamic range that can be obtained in practice with the devices, and evaluate how such image quality metrics vary across the entire field of view. Does performance degrade towards the edge of the mirror? Does performance degrade over time, as devices become de-calibrated with use?

We’ve added several components to the paper that help address the reviewer’s concerns. First, videos with additional examples are now included (Video 1), as well as images of resolution charts (Figure 1—figure supplement 2). Figure 3—figure supplement 1 has images from 8 adjacent wells (a single row) in a 96 well plate, which should give the reader a sense of how images quality and focus can vary. As the reviewer notes, image quality does degrade toward the edge of the mirror – this has now been quantified in Materials and methods: Table 2, which gives information on contrast ratio and resolution as a function distance from the optical axis. As with any system, it can be decalibrated with use, but given that we are working at relatively low magnification we have not found this to be a significant concern.

2) The manuscript is additionally weakened by the absence of a non-trivial measurement made with the device. Pilot experiments are included, demonstrating that images can be collected. However, no evidence is provided to show that these images can be used to compare samples and draw biological conclusions from them. A more convincing proof-of-principle would involve the measurement of some non-trivial biological difference between samples measured with the device, either confirming previous work or discovering something new.

We agree that our original submission was missing a substantive example. We constructed a new imaging system to address this concern and now have a convincing proof-of-principle study (see Figure 3).

3) The authors highlight the comparative simplicity of their method: it eliminates the need for motorized samples or cameras. However, this simplicity must come at some: for example a substantially increased use of space or perhaps an increase in delicate calibration required, or equipment price. If a 0.25 meter mirror is required to measure four *C. elegans* plates, how large a mirror would be required to measure 16 plates-the number that can typically be measured using a flatbed scanner? The authors could also expand greatly on other practical issues: for example, is a dedicated imaging table required to align mirrors and samples? Readers would benefit from a clearer evaluation of the practical trade-offs in deploying parabolic mirrors in a laboratory setting relative to other imaging approaches.

We thank the reviewer for this suggestion. A new section (Materials and methods: Practical considerations) has been added that addresses their concerns, and details regarding vibration isolation was added to Table 1 (we used small Sorbothane pads to reduce vibrations when needed, which is inexpensive). As we discuss in the “Practical considerations” section, the systems we designed have a small footprint and are low cost. Rather than scale the mirror, the number of systems could be scaled as total cost primarily depends on the number of imaging objectives used. Multiple systems may well be preferable as this would allow slower cameras to be used, and optical distortions introduced by large axial distances would be less evident.

Reviewer #3:The authors present a cool new idea: using a large parabolic reflector in combination with a macroscopic lens array and rapidly modulated LED array to enable fast image multiplexing between spatially separated samples. I believe that there may be interesting applications that would benefit from this capability, although the authors have not clearly demonstrated one. The paper is short, and light on discussion, details, and data.1) The manuscript does not discuss several standard, key topics for any new microscope paper: "objective" numerical aperture, image resolution, optical aberration (other than distortion, which is discussed), and camera sensor size.

We thank the reviewer for pointing out this omission on our part. We’ve included the details in Table 1.

2) Why was an array of low-performance singlet lenses used? With that selection, the image quality cannot be good. Can the system not be paired with an array of objectives or higher performance multielement lenses?

It is certainly true that the architecture could incorporate higher specification imaging objectives. However, for the 96 well systems and higher there are no commercial multi-element lenses available. Also, in terms of practicality, one of our aims was to keep costs down to a level where these systems would see widespread use and be easy to duplicate to increase total capacity. We now address this in section “Practical considerations”

3) Fluorescence imaging is not discussed or demonstrated but would obviously increase the impact of the microscope. At least some discussion would be helpful.

We agree that fluorescence would increase the impact of the microscope. However, this system is designed to be used for brightfield imaging applications. This allows us to achieve high frame rates without compromising signal to noise. Fluorescence imaging may be possible, but the low NA of the imaging objectives would severely limit the SNR and /or the rate of image capture. This issue is now also addressed in “Practical considerations”.

4) Actual HTS applications are almost always implemented in microtiter plates (e.g. a 96-well plate) to reduce reagent costs and enable automated pipetting, etc. I do not believe anyone would implement HTS in thousands of petri dishes. The paper would be strengthened substantially by a demonstration of simultaneous recording from all (or a large subset) of the wells in a 96-well plate. It's not clear whether this is possible due to the blind spot in the center of the parabolic mirror's field of view that is blocked by the camera.

We thank the reviewer for this suggestion. We now have a study that meets this criterion (Configuration 2 in Table 1, and data in Figure 3 and its corresponding video).

5) One of the primary motivations for this approach is given in the first paragraph as: "wide-field imaging systems [which capture multiple samples in one frame] have poor light collection efficiency and resolution compared to systems that image a single sample at a given time point." With a f = 100 mm singlet lens, the light collection efficiency of the demonstrated microscope is also low (estimated NA = 0.12) and the resolution is unimpressive with the high-aberration lens and 1x magnification. They demonstrated only trans-illumination applications (e.g. phase contrast), where light collection efficiency is not important. I believe a fancy photography lens mounted directly on a many-megapixel camera set to image all or part of a microtiter plate could likely outperform their system in throughput and simplicity, at least for the demonstrated applications of cardiomyocytes and *C. elegans*.

We agree that the alternative suggested by the reviewer may be viable. However, while a single-lens system could achieve similar light collection efficiency, the system would necessarily be very large (and considerably more expensive, as telecentric optics may be needed to image off-axis wells). The increased size would prohibit the use of incubators for exploring a range of sample environments. Finally, the frame rate of the machine vision camera is much higher than that of a high-resolution camera: the ssHTS system allows for fast random access capture for any sample under the parabolic mirror, allowing comparison between samples at high frame rates, which something that a conventional setup can’t do.

[Editors' note: further revisions were suggested prior to acceptance, as described below.]

Overall, the reviewers recognised that this setup could be useful for readers looking for an inexpensive bright-field imaging setup for multi-well imaging without fluorescence. They agree that you have provided substantial additional data and analysis that support your claim that parabolic reflectors can be useful for studying many samples in a parallel. The new images and videos of a 96-well plate were judged compelling. In particular, the prospect of focusing samples simply by adjusting the wavelength of illumination was thought an important step towards the goal of designing a "solid-state" imaging apparatus with no moving parts.Nevertheless, although the reviewers were satisfied that you had addressed most or all of the material points, they had considerable reservations about the way in which these improvements were presented and could not support publication of the manuscript in its present form. Indeed, there are numerous lacunae and parts where the writing is not at all clear and leads to confusion about how the system functions and what its limits are. Please find below a summary of their most important comments and a series of points made by individual reviewers, all of which would need to be addressed in a revised manuscript. If this would require a further round of experimental work, then I am afraid that we will not be able to consider your work for publication.The description of the 96-well plate data was considered both terse and vague, leaving unclear several aspects of experimental design and interpretation.– If no samples were loaded into columns 6 and 7 of the 96 well plate because of the use of 40 LED arrays, this should be stated explicitly.

We have added the following lines to the document:

In Materials and methods, Sample Preparation and Imaging:

“50 µL of this worm suspension was loaded into a 96-well, flat-bottom assay plate (Corning, Costar), excluding half of row 5 and all wells in rows 6 and 7 as shown in Figure 3A, as these wells were either obscured by sensor hardware or not illuminated by the two 40-element LED arrays (see Configuration 2 in Table 1).”

And, in Materials and methods: Optical Setup, Table 1:

“Camera placement obscures 12 wells in the 96 well plate imaged in configuration 2 (see Figure 3A), and the use of two commercial 40 element LED arrays precludes imaging all wells in a 96 well plate as the LEDs are permanently mounted on a board that is too large to be tiled without leaving gaps.”

– What was the exact reason for not imaging in column 4 of row E or row 1F.These discrepancies between the theoretically predicted function of the device and its practical performance must be clarified.If these issues do not reflect technical limitation of the device, you would need to demonstrate that these columns/wells can be imaged just like the others (i.e. this is a criterion for rejection).

In order to demonstrate to the reviewer that all wells that aren’t obscured by the camera are imageable, we present the above composite calibration image (of an overhead sheet printed with random characters with a 1.1 mm height font) which was generated by moving a single 40 element LED array and lenses to cover all locations corresponding to wells in a 96 well plate.

The system described in the paper uses two commercial 40 element LED arrays. The arrays cannot be placed side by side without gaps (see Author response image 1 – parts the board reserved for input pads are highlighted in yellow), so the use of these particular arrays precludes complete coverage of a 96 well plate without moving the array or the 96 well plate. This would not be an issue with a different array (e.g. a custom-built LED array or an LED array from another manufacturer), and we consider this a practical instead of a technical limitation. These LED arrays were chosen as they were easily sourced during the university shutdown from a Canadian supplier.

In the paper, wells E4 and F1 were obscured by a cable running from the camera. This wasn’t noticed until after the first images for the experiment shown in Figure 3 were collected, and was left in place in order to avoid moving the sample and reorienting the camera (which would have involved partial disassembly of the microscope). The orientation of the camera and cable was such so that this was not an issue when collecting the calibration image. The orientation of the cable for the experiment in Figure 3 and orientation of the cable the calibration measurement (Author response image 1) are shown in Author response image 1. As you can see, the black cable (indicated by the large red arrow) is angled so that it passes between the sample and the lenses in the left panel. As a result, the cable partially obscured wells E4 and F1 (indicated by small red arrows) in the experiments.

**Author response image 1. respfig1:** (**A**) composite image showing images from all wells aside from those under the camera housing, with wells obscured by cabling in Figure 3 highlighted in yellow. (**B**) Image of one of the LED arrays with region reserved for electrical connections highlighted in yellow; the size of these regions prevents tiling the array and complete coverage of the 96 well plate (**C**) images of the system showing the location of the cable (red arrows) that obscured the wells in Figure 3, and its orientation for the image collected in the top panel of this figure (panel A).

We have amended the text to clarify the missing wells in Figure 3 with the following three changes:

i) In the previous submission, we originally indicated that the wells were obscured in the legend of Figure 3 with the statement: “Wells obscured by hardware are denoted by an “X” symbol.”

We have changed this to read: “Wells obscured by hardware are denoted by an “X” symbol (see Materials and methods: Table 1)”

ii) And in Materials and methods: Table 1, we have added the following clarification:

**“**In addition, some wells (marked with an “X” in Figure 3a) were inadvertently obscured by hardware between the sample and objective lenses for the motion quantification experiment in Figure 3, however the number of imaged wells was considered to be sufficient to demonstrate the utility of the RAP system.”

iii) As the calibration image is useful in the context of building the same RAP system that was used to capture data in Figure 3, we include this image as part of a zip file which also includes images of the LED array and 3D printed parts with associated STL files. We also note that since author response images and text are available to *eLife* readers, the information about wells E4 and F1 will be presented in the context of the reviewer’s question and remain accessible.

The details about acquisition are so poorly described that one reviewer wrote, "why not leverage those capabilities to scan 33 wells in parallel at 15 Hz rather than one well at a time at 15 Hz?". This illustrates how you have failed to convey clearly that the system captures data from multiple wells in parallel at 120-500 fps. One video does show how 120 fps can be divided up across 80 wells, and it is illustrated in Figure 1, but these details need to be explicitly stated in the text. In Figure 2, a faster (500 fps) camera of lower resolution is used. As well as making all acquisition details clearer, you will need to provide an explicit discussion of camera choice, and any trade-off between image resolution and speed. Additionally, you need to address another technical limitation and trade-off, namely rates of acquisition and data transfer so that the possibility (and cost) of implementation in a HTS setting (see below) is clear.

We regret the confusion caused by our previous submission, and fully agree that the wording was not clear. We have amended the text in several places to make these details clear.

In the main text, the description of data acquisition was changed from:

“Motion was quantified by measuring the fraction of pixels per frame that display a change in intensity of over 25% for 100 sequential frames captured at 15 fps for each well (Figure 3A and B; Video 1).” to “… motion was quantified by measuring the fraction of pixels per frame that display a change in intensity of over 25% for 100 sequential frames captured at 15 fps/well by measuring groups of eight wells in parallel (see Figure 3A and B; Video 1, and Materials and methods: Image processing).”

The last sentence of the same paragraph has been changed from:

“As the system captures data from multiple wells in parallel, the time needed to measure activity in 76 wells for this assay is just over one minute.” to “As the system captures data from multiple wells in quick succession at a rate of 120 fps, the time needed to acquire 100 frames for each of the 76 wells for this assay is just over one minute.”

The relevant sentence in the legend in Figure 3 was changed from:

“Motion is estimated by summing absolute differences in pixel intensities in sequential frames imaged at 15 Hz.” to “wells in each row are imaged in parallel (8 wells at 15 fps per well), and net motion is estimated in each well by summing absolute differences in pixel intensities in sequential frames (see Materials and methods: Image analysis).”

We have also gone into greater details on camera choice and limitations associated with data throughput in the Materials and methods section. The following section was added:

“The camera used in Figure 2 was chosen for its high frame rate as we were interested in imaging cardiac activity, which in our experience requires 40fps acquisition speeds. […] A faster hard drive (e.g. an SSD) or RAID array would significantly increase throughput.”

The center of the optical system is intrinsically blind since space is required to position the detector. This point is implicit and must be documented as a function of the magnification.

The following sentence was added to Table 1: “Camera placement obscures 12 wells in the 96 well plate imaged in configuration 2 (see Figure 3A).”

However, we note the number of obscured wells depends only on the physical size of the camera and is independent of optical magnification.

The microscope resolution in the 15 – 20 µm range is poor relative to the sub-micron resolution of a traditional microscope. It is probably not good enough, for example, to tell individual mammalian cells apart in a confluent monolayer. This will limit the range of potential applications. Thus the spatial resolution needs to be stated in the Abstract or Introduction, not buried deep in the Materials and methods.

We have added the following line to the Introduction:

“We demonstrate the system in two low-magnification, low resolution settings using single element lenses and other easily sourced components. “

However, the use of low magnification, low NA lenses is not a defining limitation of our method. We have added the following paragraph to the Materials and methods section to clarify this point and make the reader aware of potential issues with moving to higher magnification:

“The use of low magnification optics in our current implementation is not a defining property of RAP, as higher NA, high magnification optics could be used. […] As is the case with conventional microscope designs, a high magnification RAP system would likely require a mechanism for finely adjusting objective heights to keep each sample in focus, as the depth of field of the objective lenses would be reduced.”

Further, you will need to include a detailed comparison with a standard commercial widefield microscope with a scanning stage (resolution, imaging modalities, scan time, defocusing over time, cost, integration into robotic workflows).

We have added the following text and table to the Materials and methods section, and included three additional references to support the comparison:

“While the resolution of a RAP system is similar to conventional microscopes, RAP systems differ from conventional microscopes in several respects. Table 2 summarizes some key differences between a conventional automated widefield imaging microscope and the two RAP systems implemented in this publication. We note that higher performance RAP systems (e.g. faster disks, a faster camera, corrected optics) would display improved performance.”

In addition, we have added the following to the discussion to ensure that readers are aware of the differences between our system and a conventional automated microscope:

“Automated microscopes excel at applications where data can be acquired from samples sequentially as a single high numerical aperture (NA) objective is used. […] Here, the speed increase afforded by RAP must be weighed against the many benefits of using a mature technology such as the automated widefield microscope (see Table 2 for a comparison between these systems).”

If you wish to claim HTS capacity, the comparison should also include a dedicated commercial HCS/HTS system, and the many other features needed for HTS (e.g. see https://www.ncbi.nlm.nih.gov/books/NBK558077/).Alternatively, in the absence of easy incorporation of the system in an automated setting, at a time when HTS can mean >50,000 tests/day, "High Throughput" should be removed from the title ("multi-sample" or "multi-well" would be better), and any suggestion in the text that your system is HTS-compatible seriously toned down.

We have removed high throughput from the title and removed the majority of references to high-throughput in the paper. For example, the lead sentence in the Introduction now reads:

“Conventional multi-sample imaging modalities either require movement of the sample to the focal plane of the imaging system ^1–4^, movement of the imaging system itself ^5,6^, or use a widefield approach to capture several samples in one frame ^7,8^.”

The term “high-throughput” is still occasionally used, but its context is limited to imaging multiple samples rapidly, or when discussing other platforms. Table 2 also states that the RAP system hasn’t been validated as part of a conventional high-throughput workflow.

Equally, given the very different uses in optics or electronics of the term "solid-state", you should avoid it in the title, replacing it, for example by, "with no moving parts".

We have removed the term solid-state from the title as requested.

There was also a general consensus that your design is not a Newtonian telescope, which has two mirrors instead of a single mirror as in this design. The reviewers recommend changing "novel Newtonian telescope design" to "large on-axis parabolic mirror design", "parabolic reflector", or something similar that is clearer and more accurate. Including a phrase like "inspired by a Newtonian telescope" would be acceptable.

We now use the phrase “inspired by Newtonian telescope” instead of “modified Newtonian telescope” as requested.

Further points made by individual reviewers:1) The authors compare wild-type *C. elegans* to nuo-6 mutants. The authors are vague and qualitative in their descriptions of movement. Nuo-6 mutants are predicted to "move less frequently" than wildtype. This is confusing, as *C. elegans* generally exhibit some degree of continuous movement as long as they remain alive, involving body postural changes, head movements, or pharyngeal pumping. Are the authors referring to the frequency of a particular type of movement? For the purposes of this paper, the authors probably do not need to alter their imaging pipeline, but they should be substantially more specific about which behavior their method is measuring.

We regret the confusion and have amended the text as follows:

“We validate the potential for RAP to be used in a higher-throughput imaging application by measuring motion in *C. elegans* mitochondrial mutant nuo-6(qm200)18, which have a slower swimming rate (frequency of thrashing) than that of the wild type *C. elegans*. […] As the system captures data from multiple wells in quick succession at a rate of 120 fps, the time needed to acquire 100 frames for each of the 76 wells for this assay is just over one minute. “

The motion estimate we use is not based on the conventional method used to quantify thrashing frequency in *C. elegans*, but rather quantifies pixel intensity changes between frames.

We stress that our intention was not to measure thrashing frequency differences in *C. elegans* wild type and mutant strains, as this is already known. Rather, we leverage the fact that the documented difference in thrashing frequency will generate a measurable difference when quantifying pixel intensity changes in wells containing these two strains. Our aim was (only) to use these strains to demonstrate that the system can quantify differences in activity multiple, simultaneously imaged wells.

We have added the following statement (in Materials and methods: Image processing) in order to further clarify our intent:

“We note that while this algorithm yields results which are consistent with published manual measurements of thrashing frequency (see Figure 2j in Yang and Hekimi^18^), there is no direct correspondence between this metric and specific behaviours (head movement, posture changes etc.). However, the documented difference in the activity of the two strains we use would predict the difference in the metric that we observe and can be used as a validation of the imaging method to track movement over time.”

2) Many nematode behaviors change in response to stimulation with light, physical stimulus, or immersion in liquid. Other behaviors are suppressed by long periods spent immersed in un-mixed liquids. It remains difficult to interpret the authors' results without additional information describing how the light and culturing conditions they are use influences nematode behavior and how this influences their results. In particular, the behavioral difference observed between day 1, 2 and 3 could be expected as a technical artifact (i.e., in the absence of any underlying aging process) if nematodes remained in the same wells for multiple days.3) The authors observe a difference in activity between nuo-6 and wild-type animals, and also between young animals and old animals. However, discussion of this is surprisingly qualitative given the quantitative thinking found elsewhere in the paper. Are the observed differences in movement approximately the same magnitude as what would be expected given previous results? Why is a significant difference between the two strains observed only on day one and three, but not day two?

We agree that the inclusion of the two extra imaging days is a potential source of confusion as animals are impacted by multiple inputs, and so we have opted to remove the data for days 2 and 3. The remaining data (on day 1), along with the included video examples are sufficient to demonstrate that we can use the system to collect data from multiple wells, which is the focus of the submitted paper.

4) The Figure 1 caption uses fM and fL while Figure 1 uses F1 and F2. Please make consistent.

We thank the reviewer for pointing this out. We have edited the figure to be consistent.

5) Equation 2 is not fully displayed.

We thank the reviewer for pointing this out. It was cropped during the pdf conversion process, and we will ensure that it is displayed properly in the submitted version.

6) Introduction: Please give some concrete examples of experiments that require continuous long-term recording where low-resolution brightfield imaging would be the appropriate readout modality.

We have added the following text to the discussion, as well as four references:

“RAP systems are better suited for dynamic experiments in relatively macroscopic samples where multiple continuous long-duration recordings are the primary requirement. For example, rhythms in cultured cardiac tissue evolve over hours^28^ or even days^29,30^, but display fast transitions between states (e.g. initiation or termination of re-entry^31^), necessitating continuous measurement. In these experiments, moving between samples would result in missed data.”

7) Introduction: The phrase "high resolution" is misleading, as the 15-22 µm resolution of this microscope would be considered very low resolution by most microscopists. Please insert the actual resolution here.

We originally intended for high resolution to refer to pixel count not numerical aperture but agree that this is confusing. We have removed the term.

8) Results: I would not call this a high light collection efficiency design, as most standard microscopes have higher efficiency. Light collection efficiency is not very important here, so please change the language to be less contentious.

The language has been changed to: “The brightfield nature of the illumination used in this design allows images to be captured with sub millisecond exposure times. “

9) Results: Calling an LED source spatially coherent is really straining the definition. Please use different language.

While the reviewer is right to state that LEDs are not normally coherent sources, they can be coherent if the emitting area is small. The following images (Author response image 2), taken at the same magnification, demonstrate that the emitting areas of these LEDs is approximately 200 x 200 um. This provides a spatial coherence value > 0.5 compared to a spatial coherence value of 0.88 for a DPSS laser (e.g. see DOI:10.1038/s41598-017-06215-x). It is therefore safe to assume that these emitters have a high degree of spatial coherence.

**Author response image 2. respfig2:** LED emitting area is approximately 200x200 microns.

We have added the reference to the paper. We have also changed the text from

“coherent” to “partially coherent” to avoid confusion:

“The illumination of the sample by a spatially partially coherent source^12^ produces grey scale images, and in our studies, it is the change in this intensity that is of interest. “

10) Materials and methods: Something is wrong or confusing about the depth of focus discussion. Please cite a source for the equations and clearly define all variables. If u = f as you indicate in the text, then DOF=2c≠0.9 mm, which was stated in the text. The f-number does not appear in the equation you have, but the discussion seems to indicate that it is important (as would be expected).

We thank the reviewer for spotting this error. There was a typo in the equation, and the f-number is now included.

11) Figure 2 legend: should be "(blue trace in B)"

We thank the reviewer for spotting this typo. The legend has been changed.

12) Figure 3 legend: duplicated text "C) Focal plane…."

We thank the reviewer for spotting the duplicate text- it has been removed.

13) The authors limit their discussion of statistical analysis of animal movement to the legend of Figure three. This analysis would seem more natural to include either in the main text or in a dedicated statistical methods section

The following line is now in the main text:

“Instead of measuring thrashing frequency directly, motion was quantified by measuring the fraction of pixels per frame that display a change in intensity of over 25% for 100 sequential frames captured at 15 fps/well by measuring groups of eight wells in parallel (see Figure 3A and B; Video 1, and Materials and methods: Image processing).”

In addition, there is now new section (Image Processing) in Materials and methods with added details:

“Image processing: We find that image brightness drops with increased objective lateral distance and that images are subject to aberrations at the edges. […] The values are plotted as a percentage.”

14) Provide more precise references to allow others to set up an ssHTS system; see for example the references for LEDs.

We thank the reviewer for this suggestion. The LED array used (manufacturer and part name) is now in Table 1. In addition, as mentioned earlier in this response, we now include a zip file with additional details (including part numbers and stl files) targeted to readers interested in building their own device.

[Editors' note: further revisions were suggested prior to acceptance, as described below.]

The manuscript has been improved but there are some remaining issues that need to be addressed before acceptance, as outlined below:The authors stress that one of the principal interests of the system is the capacity for rapid and continuous imaging. They write, "captures data 15 fps/well by measuring groups of eight wells in parallel". Then they write, "As the system captures data from multiple wells in quick succession at a rate of 120 fps, the time needed to acquire 100 frames for each of the 76 wells for this assay is just over one minute". They need to be more explicit. When they are capturing data from 76 wells, then are they imaging each well at ca. 1.5 fps? As it stands, a reader might understand that they are switching between groups of eight wells, imaging one group at 15 fps/well, then moving to the next group after capturing 100 frames (6.7 seconds). If this were the case, then they would return to image the first group after a minute, so their system would not be continuous. This clearly requires clarification.

We apologize for the confusion. As shown in Video 1, we can indeed image all wells in parallel continuously at 1.5 fps/well. However, for Figure 3, in order to capture 15 fps per well (which was needed to visualize *C. elegans* thrashing behaviour) we capture a subset of wells in parallel and switch to different subsets (every 6.7 seconds) until all the wells are imaged. The entire data set is 100 sequential frames for each well, so we do not return to the first group for this particular assay. We have changed the text in that paragraph as follows:

In this experiment, the frame rate of the camera is limited to 120 fps (see Materials and methods: Practical considerations and Video 1), allowing us to image 8 wells in parallel at 15 fps/well. 80 wells (76 active and 4 blank wells – see Figure 3A) are imaged by measuring 100 frames from each well in a row of 8 wells in parallel (800 frames/row) before moving to the next row, until all 80 wells are imaged (a total of 8000 frames). The system quantified decreased activity in *nuo-6(qm200)* which is consistent with published results18 (Figure 3B). The time needed to perform this assay is just over one minute (8000 frames/120 fps = 67 seconds).

We also note that relationship between framerate, data throughput and the number of samples that can be imaged continuously in parallel is discussed in more detail in “Materials and methods: Practical considerations.”